# Construction of Development Scores to Analyze Inequalities in Childhood Immunization Coverage: A Global Analysis from 2000 to 2021

**DOI:** 10.3390/ijerph22060941

**Published:** 2025-06-16

**Authors:** Andrea Maugeri, Martina Barchitta, Syed Muhammad Zaffar, Antonella Agodi

**Affiliations:** 1Department of Medical and Surgical Sciences and Advanced Technologies “GF Ingrassia”, University of Catania, Via S. Sofia 87, 95123 Catania, Italy; andrea.maugeri@unict.it (A.M.); martina.barchitta@unict.it (M.B.); 2Department of Economics and Business, University of Catania, Corso Italia 55, 95129 Catania, Italy; syed.md.zafar093@gmail.com

**Keywords:** vaccine, immunization, childhood, development, socio-economic factors

## Abstract

Immunization coverage is a key public health indicator reflecting healthcare accessibility and socio-economic conditions. This study employs Principal Component Analysis (PCA) to construct composite development scores and analyze their relationship with immunization coverage for measles and diphtheria-tetanus-pertussis (DTP) vaccines across 195 countries (2000–2021). The analysis comprises a training period (2000–2015) for score development and a test period (2016–2021) for validation. Variables were selected based on correlation with immunization coverage and standardized before PCA extraction. PC1, the principal component explaining the largest variance, was identified as a key indicator of development disparities. Findings reveal that higher PC1 scores (lower socio-economic development) are associated with reduced immunization rates, while lower PC1 scores (higher socio-economic development) correspond to greater coverage, a trend consistent across both periods. Geospatial analysis highlights stark disparities, particularly in sub-Saharan Africa and South Asia, whereas North America, Europe, and East Asia maintain significantly higher coverage. These results provide policy-relevant insights, demonstrating the utility of PCA-derived scores for resource allocation and targeted interventions.

## 1. Introduction

Immunization is one of the most effective public health interventions, saving millions of lives annually and significantly reducing the morbidity and mortality associated with vaccine-preventable diseases [1,2]. Despite global advancements in vaccine accessibility, disparities in immunization coverage persist and remain a significant challenge, particularly in low- and middle-income countries (LMICs) [3,4,5]. These disparities are not merely a reflection of logistical or policy shortcomings but are deeply rooted in socio-economic inequalities that shape access to healthcare and vaccination services [5].

Socio-economic factors, such as income inequality, education levels, healthcare infrastructure, and technological access, play a critical role in determining immunization coverage [6,7]. Countries with higher socio-economic development tend to have robust immunization programs, resulting in higher vaccine coverage rates [4,5]. Conversely, countries with limited resources and weaker healthcare systems face substantial barriers to achieving equitable vaccine distribution [8]. This dichotomy is evident in regional and global patterns, where vaccination coverage is consistently lower in LMICs compared to high-income countries [4,5]. Such disparities underscore the importance of addressing the socio-economic determinants of immunization to achieve global health equity.

Efforts to address these challenges have been guided by international initiatives, such as the United Nations Sustainable Development Goals (SDGs) and the Immunization Agenda 2030 [9,10]. The SDGs emphasize reducing health inequities and ensuring access to essential healthcare services for all populations [9], while the Immunization Agenda 2030 focuses on increasing equitable access to vaccines [10]. The latter, in particular, seeks to halve the number of children missing critical vaccines and improve vaccination rates in under-immunized communities [10]. However, these goals face substantial hurdles, particularly in the wake of disruptions caused by the COVID-19 pandemic. In fact, the pandemic worsened pre-existing disparities, making it even more evident that tailored strategies are needed to overcome the socio-economic barriers to immunization [11,12].

The disparities in immunization coverage highlight underlying inequalities in socio-economic development. For example, routine childhood vaccine coverage varies significantly between countries, with those having higher Gross Domestic Product (GDP) per capita typically achieving greater coverage than those with lower GDP per capita [5]. However, while GDP per capita offers important insights into economic performance, it falls short of reflecting the complex and multi-dimensional aspects of development that influence public health outcomes. Factors, such as education, healthcare quality, infrastructure, and demographic dynamics, play equally significant roles in shaping public health outcomes, including immunization coverage [6,7,13,14,15,16]. The use of composite metrics that integrate these diverse dimensions offers a more comprehensive framework for understanding the socio-economic determinants of immunization coverage [17,18].

Building on this need, composite scores can provide a nuanced understanding of how different aspects of development influence vaccination coverage. Such metrics can help identify gaps in immunization coverage that may not be immediately apparent through conventional economic indicators. This study aims to address these gaps by constructing composite development scores using Principal Component Analysis (PCA), a robust statistical technique for dimensionality reduction [19,20,21]. By leveraging data from 195 countries over a 21-year period (2000–2021), this study captures both spatial and temporal patterns in childhood immunization coverage for measles and diphtheria-tetanus-pertussis (DTP). These two vaccines were selected as they serve as key indicators of national immunization program performance and broader healthcare system effectiveness. Measles vaccination is crucial due to its high transmissibility and the need for herd immunity to prevent outbreaks, while DTP vaccination is a widely monitored measure of routine childhood immunization, reflecting accessibility and delivery efficiency of primary healthcare services [4,5].

The analysis focuses on two main objectives: first, to construct socio-economic development scores that encapsulate the multi-dimensional nature of socio-economic progress; and second, to examine the relationship between these scores and immunization coverage. This dual approach offers a unique perspective on the factors driving disparities in childhood immunization coverage and provides a framework for identifying areas where targeted interventions are most needed.

## 2. Materials and Methods

### 2.1. Study Datasets

The dataset utilized in this study combines publicly available, globally recognized databases to examine the relationship between socio-economic development and immunization coverage. The primary data sources include the World Bank DataBank [22] for World Development Indicators and the UNICEF Data Warehouse [23] for immunization coverage statistics. All relevant details regarding data origin, collection processes, and the methodologies supporting these datasets are documented in the official sources and accompanying dataset documentation [22,23]. These datasets were integrated to construct a comprehensive longitudinal dataset, where each observation corresponds to a specific country in a given year. Spanning 195 countries over a 21-year period (2000–2021), the final dataset comprises 4290 country-year observations.

### 2.2. Study Variables

The primary dependent variables are immunization coverages for measles and DPT, expressed as the percentage of children aged 12–23 months who received the respective vaccines [23]. These indicators serve as critical measures of immunization program success and equity across countries and over time. In addition to the dependent variables, the dataset comprises 158 World Development Indicators that span a wide array of domains [22] as follows: health-related indicators include measures, such as life expectancy at birth, maternal mortality ratios, age-specific mortality rates, and health expenditure patterns; economic indicators capture dimensions like GDP per capita, employment distribution, and financial resources allocated to health; education and societal factors encompass metrics, such as compulsory education duration, the proportion of parliamentary seats held by women, and school enrolment rates; infrastructure and technological development are assessed through variables, such as access to electricity, internet penetration, and mobile cellular subscriptions. Additionally, demographic dynamics are represented by indicators, such as adolescent fertility rates, population density, and urban population growth. A complete list of these indicators is available in the Appendix A for reference.

### 2.3. Data Management

To ensure the reliability of analyses, a systematic data management protocol was employed to address missing values. As an initial pre-processing step, rows and columns with more than 25% missing values were excluded, leading to the removal of 549 country-year observations and 110 indicators. A comprehensive summary of the remaining missing data by indicator is provided in the Appendix A, enabling a transparent evaluation of data coverage and reliability. For the remaining dataset, missing values were imputed using a combination of forward and backward methods, aimed at minimizing potential biases while preserving the overall integrity of the time series structure. Following this procedure, the final analytical dataset comprised 3741 country-year observations and 48 indicators, in addition to the measles and DPT immunization coverage variables. Summary statistics for these indicators are reported in the Appendix A.

### 2.4. Construction of PCA-Based Scores

To explore the relationship between socio-economic development and immunization coverage, a data-driven methodology was employed to construct two composite scores encapsulating the multi-dimensional nature of socio-economic development. These scores were generated using PCA, a robust statistical technique designed to reduce the complexity of large datasets while retaining the maximum possible variance and underlying structure. To ensure methodological robustness and assess the predictive validity of the constructed scores, the dataset was divided into two distinct periods: a training period (2000–2015) and a test period (2016–2021). This division was selected based on key considerations as follows: the 15-year training period provides a sufficiently large dataset for PCA to identify stable and meaningful patterns, reducing the risk of overfitting and ensuring that the extracted principal components (PCs) effectively capture long-term socio-economic trends; the subsequent 5-year test period was selected to assess whether the PCA-derived development scores remain consistent when applied to new, unseen data. This approach ensures that the scores are not limited to the training period but can reliably generalize to future data [19].

The analysis began with the selection of a subset of indicators from the 48 socio-economic and health-related variables included in the dataset for the training period. As there is no prior established knowledge regarding which specific indicators are most strongly associated with immunization coverage, a purely data-driven approach was adopted. The selection criteria were determined based on the strength of the association between each indicator and immunization coverage for measles and DTP. Spearman’s rank correlation coefficients were computed to quantify these associations, given their suitability for assessing monotonic relationships. Indicators with statistically significant correlations (*p* < 0.05) were selected, applying a threshold of an absolute correlation coefficient > 0.3. This threshold was chosen to strike a balance between inclusivity and relevance—while a higher cut-off (e.g., 0.5 or 0.7) would capture only the strongest associations, it could inadvertently exclude socio-economic variables that play an indirect yet meaningful role in immunization disparities [19]. At this stage, no adjustment for multiple comparisons was applied, as the purpose of this analysis was not to establish definitive statistical significance but rather to identify variables for inclusion in the PCA, ensuring a comprehensive representation of socio-economic determinants in the dimensionality reduction process [24]. This approach aligns with best practices in dimensionality reduction, where moderately correlated variables can collectively enhance the explanatory power of principal components [25].

PCA was then applied to identify latent dimensions of socio-economic development by capturing the most significant sources of variation within the dataset. This technique enables the derivation of composite scores that encapsulate multiple aspects of socio-economic development in a structured and interpretable manner. To ensure comparability across indicators with different measurement scales, z-score standardization was applied. For each year, all variables were transformed to have a mean of zero and a standard deviation of one, maintaining relative differences between values while ensuring that all indicators contributed equally to the PCA model. As is standard practice in PCA, this transformation prevents variables with larger scales from disproportionately influencing the results. The use of z-score normalization is widely recognized in statistical and machine learning applications, as it enhances comparability and ensures that the extracted PCs capture meaningful variance across all indicators rather than being dominated by high-magnitude variables [19,25]. For factor extraction, Varimax rotation was applied, an orthogonal rotation method that enhances interpretability by maximizing the variance of squared loadings for each principal component. This method was selected because it maintains the independence of components, ensuring that the resulting factors capture distinct and non-overlapping dimensions of socio-economic development [19,25].

PCA was applied to the training period dataset to extract a set of mutually orthogonal PCs, which are linear combinations of the original variables designed to maximize variance capture. These components are hierarchically ordered based on the proportion of total variance they explain. The first component (PC1) captures the largest share of variance, representing the dominant underlying dimension in the data, while subsequent components (PC2, PC3, etc.) explain progressively smaller amounts of variance, orthogonal to the preceding components. The relationship between each original variable and a PC is quantified by a factor loading, a coefficient that measures both the strength and direction of the association. A high absolute loading value indicates that the variable strongly influences the respective PC, with positive and negative loadings reflecting the direction of this influence. These loadings provide critical insights into the structure of the data, revealing patterns and clusters of variables that share similar behaviors or contributions to specific components.

In this way, two composite scores were derived from the principal components identified during the training period. The PC1 score corresponds to the first PC1, which captures the single largest source of variance in the dataset and represents the dominant dimension. The PC1 score for the *i*th country-year observation is calculated as follows:$${PC1\;Score}_{i}=\sum_{j=1}^{n}{Loading}_{j,1}\cdot {Variable}_{i,j}$$
where:

${Loading}_{j,1}$ represents the factor loading of the *j*th variable on PC1,${Variable}_{i,j}$ represents the value of the *j*th variable for the *i*th country-year observation,*n* is the total number of variables included in the analysis.

The Overall PC Score aggregates all principal components (PCs) required to explain a minimum of 90% of the total variance, thereby providing a multi-dimensional and integrative representation of a country’s socio-economic profile. The selection of this cumulative variance threshold ensures a comprehensive depiction of socio-economic development across diverse dimensions, enhancing the robustness of the analysis. Rather than adhering strictly to the Kaiser criterion (i.e., retaining only components with eigenvalues greater than one), additional components with eigenvalues marginally below this threshold were intentionally included. The overall PC score for the *i*th country-year observation is calculated as follows:$${Overall\;PC\;Score}_{i}=\sum_{k=1}^{m}\sum_{j=1}^{n}{Loading}_{j,k}\cdot {Variable}_{i,j}$$
where:

${Loading}_{j,k}$ represents the factor loading of the *j*th variable on the *k*th PC,${Variable}_{i,j}$ represents the value of the *j*th variable for the *i*th country-year observation,*m* is the number of PCs required to explain at least 90% of the total variance,*n* is the total number of variables included in the analysis.

To assess the stability and applicability of the PCA model over time, we conducted an empirical comparison of PC structures between the training period (2000–2015) and the test period (2016–2021). First, we analyzed the factor loadings of the principal components and observed a high degree of similarity across both periods. Next, we compared the proportion of total variance explained by the top components, confirming that the variance distribution remained stable. Additionally, we calculated the correlation between PCA-derived scores from the training and test periods, obtaining a high correlation coefficient (0.985 for PC1), demonstrating strong consistency. These validation steps confirm that PCA-derived scores serve as reliable indicators for examining socio-economic differences, reinforcing the model’s robustness for long-term trend analysis. Consequently, we applied the PCA-derived factor loadings from the training period (2000–2015) to the test period (2016–2021) to generate consistent scores for out-of-sample data. By extending the analysis beyond the initial training period, this approach establishes a robust framework for longitudinal and comparative studies, enabling an assessment of how socio-economic development trends impact immunization coverage over time while ensuring methodological consistency.

### 2.5. Statistical Analyses

All analyses were conducted using Python (version 3.4), leveraging its statistical and visualization libraries. Descriptive statistics were computed for all variables, selecting appropriate metrics based on data distribution. Continuous variables were summarized using the mean, median, standard deviation, and interquartile range, while categorical variables were described through frequencies and percentages.

To examine the relationship between development scores (PC1 Score and Overall PC Score) and immunization coverage (measles and DPT), both visual and statistical methods were employed. Scatter plots were generated to explore continuous relationships, incorporating polynomial trend lines and 95% confidence intervals to identify potential non-linear associations. Additionally, box plots were constructed after stratifying development scores into quartiles, enabling comparisons across different levels of socio-economic development. Each year, countries were grouped into quartiles based on their PC1 Score and Overall PC Score, ensuring an equitable distribution of observations across four distinct development levels. This classification was performed post-PCA score derivation and applied consistently across both the training and test periods.

To assess differences in immunization coverage across development quartiles, the Kruskal–Wallis test was conducted. When significant differences were detected, Dunn’s post hoc test with Bonferroni correction was applied to adjust for multiple comparisons (corrected significance level = 0.008). Additionally, the association between development quartiles and immunization coverage was evaluated using confusion matrices, followed by the Chi-Square test to determine statistical significance. Agreement between categorical classifications was further assessed using Weighted Cohen’s Kappa, accounting for partial mismatches in classification. Geospatial analyses were conducted to visualize the distribution of development scores and immunization coverage quartiles by country for the year 2021, representing the most recent data available at the time of analysis. These maps were generated using Python’s geopandas library, illustrating quartiles of PC1 Score, measles immunization coverage, and DPT immunization coverage. To evaluate the robustness of the PCA-based framework, we conducted sensitivity analyses under two alternative conditions: (i) using a complete-case dataset excluding imputed values, and (ii) reducing the number of retained components from ten to six, in alignment with the Kaiser criterion (eigenvalues > 1). In both scenarios, the resulting PC scores were compared to those from the main analysis.

Unless otherwise specified, a significance threshold of 0.05 was applied to all statistical tests.

## 3. Results

The initial phase of our analysis involved computing a correlation matrix for the variables included in the dataset. This procedure was critical for feature selection and for assessing the interrelationships among the variables. Specifically, Appendix A presents a heatmap of the correlation matrix, delineating the associations between immunization coverage for measles and DTP and the 48 socio-economic development indicators under investigation. In general, both positive and negative correlations were observed. More specifically, 32 development indicators exhibited significant correlations with either measles or DTP immunization coverage, with absolute correlation coefficients exceeding 0.3. For measles immunization coverage, the most prominent positive correlations, as follows, were observed with: People using at least basic sanitation services (rho = 0.62); Access to electricity (rho = 0.61); Life expectancy at birth (rho = 0.61); People using at least basic drinking water services (rho = 0.60); Conversely, the strongest negative correlations were identified with Neonatal mortality rate (rho = −0.64) and Infant mortality rate (rho = −0.64). A similar correlation pattern was observed for DTP immunization coverage, reflecting shared socio-economic and health-related determinants with measles immunization. Additionally, GDP per capita, a widely recognized measure of economic development, demonstrated moderate positive correlations with immunization coverage, with correlation coefficients of 0.52 for measles and 0.55 for DTP. The 32 indicators meeting the correlation threshold were included in subsequent analyses to further explore their influence on immunization coverage. An additional observation from the correlation analysis is the significant intercorrelations among the socio-economic indicators (Appendix A), which aligns with expectations given their inherent dependencies. This confirms the suitability of using PCA as a dimensionality reduction technique, allowing for the extraction of key underlying factors that explain these associations effectively.

Accordingly, PCA was conducted on the training period dataset (2000–2015), resulting in the identification of ten PCs that together accounted for at least 90% of the total variance. PC1 emerged as the dominant component, explaining 57.1% of the variance, underscoring its central role in capturing the primary patterns underlying socio-economic development indicators (Appendix A). Although the scree plot (Appendix A) exhibited a distinct elbow after the first component—reflecting its predominance—we retained the first ten components for the computation of the Overall PC Score. This decision sustains a more comprehensive representation of the complex, multi-dimensional nature of socio-economic development. While the first six components met the Kaiser criterion (eigenvalues > 1), we deliberately included four additional components with slightly lower eigenvalues, recognizing their capacity to encapsulate subtler, yet potentially policy-relevant, aspects of variance that would otherwise be excluded under more restrictive criteria.

The contributions of individual variables to the selected components, expressed as loadings, are presented in Appendix A. These loadings reflect how each indicator influences the PCs, facilitating a more comprehensive interpretation of the multi-dimensional structures identified through PCA.

Next, two composite scores were constructed to quantify development based on the results of PCA. The first score was derived exclusively from PC1 (PC1 score), capturing the dominant dimension of variance in the dataset. The second score incorporated all 10 components that cumulatively explained 90% of the variance (Overall PC score), providing a broader measure of socio-economic development. Both scores were inversely correlated with GDP (rho = 0.65 and 0.62, respectively), indicating that higher scores align with lower socio-economic development. Notably, the alternative PC scores obtained in both sensitivity scenarios showed strong and statistically significant correlations with those from the main analysis (*p*-values < 0.001). This indicates that the core results are robust to variations in the imputation approach and the number of components retained.

The relationship between PC scores and immunization coverages was initially visualized using scatterplots (Figure 1), where each point represents a country-year observation and is color-coded by year. Country-year observations with low PCA scores (indicating higher socio-economic development) cluster in the upper left of the plots, where immunization coverage is highest. Conversely, as the PCA scores increase—reflecting lower socio-economic development—immunization coverages decline in a non-linear fashion. For clarity, the graphs display the trend lines for the overall sample; however, the same negative relationship is evident when examining the distribution of points for each specific year. This analysis underscores the strong link between lower socio-economic development and reduced vaccination rates for both measles and DPT.

Countries were grouped into quartiles (Q1 to Q4) based on their annual PC scores, with Q1 indicating the highest socio-economic development and Q4 the lowest. Figure 2 illustrates these differences using box plots, which effectively summarize the central tendency, variability, and outliers of immunization coverage for both measles and DPT. The figure reveals a clear, non-linear decline in coverage from Q1 to Q4. For instance, when grouped by PC1 Score, the median measles immunization coverage decreased from 94.0% in Q1 (IQR = 5.9) to 65.0% in Q4 (IQR = 20.0), while median DPT coverage declined from 95.3% in Q1 (IQR = 4.9) to 66.0% in Q4 (IQR = 24.0). Similarly, when grouped by Overall PC Score, the median measles coverage dropped from 94.2% in Q1 (IQR = 6.0) to 67.9% in Q4 (IQR = 21.9), and median DPT coverage fell from 95.0% in Q1 (IQR = 5.9) to 69.2% in Q4 (IQR = 26.0). A Kruskal–Wallis test confirmed a highly significant association (*p* < 0.0001) between development quartiles and immunization coverage. Furthermore, Dunn’s post hoc comparisons with Bonferroni correction revealed marked disparities as follows: countries in Q4 exhibited significantly lower coverage than those in Q3, Q2, and Q1 (*p* < 0.0001 for both vaccines), while countries in Q3 also showed lower coverage than those in Q2 and Q1 (*p* < 0.005 for both vaccines).

These results reveal the utility of the PCA-derived scores in capturing disparities in immunization coverage driven by multi-dimensional developmental factors. To further evaluate this relationship, Figure 3 presents confusion matrices comparing quartile-based classifications of socio-economic development (using the PC1 Score and the Overall PC Score) with quartiles of immunization coverage for measles and DTP. Each matrix cell indicates the number of country-year observations assigned to a given combination of socio-economic development score quartile and immunization coverage quartile, with darker shades representing higher frequencies. The diagonal running from the top-right to the bottom-left cell highlights concordance between a country’s socio-economic development score classification and its immunization coverage quartile. Notably, the greatest agreement occurs between Q1 of the development score and Q4 of immunization coverage, as shown by the largest frequency of observations in that diagonal cell.

A Chi-square test revealed a highly significant association (*p* < 0.0001) across all matrices, confirming the strong link between socio-economic development and vaccination coverage. Weighted kappa coefficients provided an additional measure of agreement, indicating moderate concordance between development quartiles and coverage quartiles. For the PC1 Score, these values reached 0.622 for measles and 0.653 for DTP, while the Overall PC Score yielded slightly lower coefficients (0.564 for measles and 0.554 for DTP). These results underscore the ability of PCA-derived scores to capture multi-dimensional developmental factors that drive disparities in immunization coverage, with PC1 showing a marginally stronger relationship.

Recognized as the most suitable metric, the PC1 Score was selected for further analysis. To assess its performance and temporal stability, the PCA-derived loadings were applied to compute scores for country-year observations beyond the training period, covering the years from 2016 to 2021. The analysis revealed a consistent and statistically significant inverse relationship between the PC1 Score and immunization coverage for both measles and DPT. Spearman’s Rank Correlation Coefficients for measles ranged from −0.55 in 2016 to −0.62 in 2020, while for DPT, correlations followed a similar pattern, ranging from −0.56 in 2016 to −0.60 in 2021. These results indicate that higher PC1 scores, indicative of lower socio-economic development, are consistently associated with lower immunization coverage across the test period. Furthermore, Chi-square tests confirmed a strong association between PC1 score quartiles and immunization coverage quartiles (*p* < 0.0001). Weighted Kappa statistics further demonstrated moderate agreement, reinforcing the stability and reliability of the PC1 Score in capturing disparities in immunization coverage over time, with values ranging from 0.52 to 0.57 for both measles and DPT. A comprehensive summary of the correlation, association, and agreement between the PC1 Score and immunization coverage for measles and DPT from 2016 to 2021 is provided in the Appendix A.

To demonstrate the practical application of the analysis, Figure 4 presents quartile maps of the PC1 score and immunization coverage for measles and DPT in 2021. The year 2021 was selected as it represents the most recent year for which data were available at the time of analysis. A list of countries according to PC1 score quartile distribution for 2021 is available in the Appendix A. A clear spatial relationship emerges from these maps, highlighting the inverse association between PC1 scores and vaccination coverage. Countries with lower PC1 scores, indicative of higher levels of socio-economic development, tend to exhibit higher immunization rates, while regions with higher PC1 scores, reflecting lower socio-economic development, show reduced vaccination coverage. Notably, regions, such as North America, Europe, and parts of East Asia, are consistently categorized in the lower quartiles of the PC1 score, aligning with the upper quartiles of immunization coverage. Conversely, many countries in sub-Saharan Africa and South Asia are positioned in the higher quartiles of the PC1 score and correspond to lower quartiles of immunization coverage.

## 4. Discussion

This study investigates the relationship between socio-economic development and immunization coverage, demonstrating the utility of PCA-derived composite scores in assessing global disparities. Specifically, immunization coverage for measles and DTP was examined as a key health indicator [26,27], representing both the accessibility of healthcare services and the broader socio-economic context. The findings highlight the pivotal role of various development dimensions in influencing public health outcomes, emphasizing that disparities in vaccination coverage are deeply rooted in underlying socio-economic inequalities.

A recent review underscores how the drivers of inequalities in childhood immunization coverage differ significantly across contexts [6]. In low- and middle-income countries (LMICs), economic status and maternal education are often the most critical dimensions of inequality, reflecting systemic challenges, such as limited access to education and healthcare infrastructure [28,29]. Maternal education, in particular, is a key determinant of vaccination uptake, as caregivers with higher educational attainment are more likely to understand the benefits of immunization and navigate healthcare systems effectively. Conversely, families in lower-income settings, especially those engaged in precarious or informal employment, often encounter financial constraints and logistical obstacles to accessing vaccines [28,29].

Access to healthcare infrastructure is indeed another crucial factor influencing immunization rates [30,31]. Regions with well-established healthcare systems, sufficient numbers of healthcare professionals, and reliable cold-chain facilities tend to achieve higher vaccination coverage [32]. Urban areas generally benefit from centralized healthcare facilities, but marginalized urban populations may still experience low immunization rates due to socio-economic disadvantages [28]. In rural and sparsely populated regions, logistical difficulties and geographic barriers further exacerbate inequities, as limited healthcare infrastructure complicates vaccine delivery [28]. Political stability and governance also play a significant role. Stable governments are better equipped to enforce policies, such as mandatory vaccination laws and provide financial incentives to improve coverage [33]. Conversely, regions affected by political instability or conflict often experience interruptions in immunization programs, leading to significant disparities [26,33]. In high-income countries, the primary drivers of inequality often differ, with systemic biases and structural discrimination based on race and ethnicity emerging as key factors [34]. These inequities highlight the distinct pathways through which social, economic, and structural determinants influence vaccination coverage across different settings, emphasizing the need for tailored interventions to address the unique challenges faced by each context [35,36]. This context-specific understanding stresses the importance of employing metrics like PCA-derived composite scores, which integrate multiple dimensions of development to offer a more comprehensive perspective on disparities. Unlike traditional indicators, such as GDP per capita, which provide a narrow focus on economic performance [6], these scores encapsulate a broader range of socio-economic, health-related, and infrastructural factors.

The application of PCA-derived loadings from the training period (2000–2015) to country-year observations between 2016 and 2021 demonstrated the stability and predictive utility of the composite development scores. The consistent negative associations between these scores and immunization coverage throughout this period, reflected in stable Spearman’s correlation coefficients and moderate weighted kappa values, validate their robustness as proxies for socio-economic disparities. Notably, the stronger performance of the PC1 score compared to the Overall PC score underscores the critical role of the dominant dimension of socio-economic variability in shaping immunization outcomes, while highlighting the complementary, albeit less pronounced, influence of additional dimensions. The geographic disparities identified in this analysis further contextualize the relationship between socio-economic development and immunization coverage. Countries with higher PC1 scores, indicative of lower levels of socio-economic development, continue to face significant barriers to achieving equitable immunization coverage. These barriers often include insufficient healthcare infrastructure, financial limitations, and restricted access to education, which collectively hinder vaccination efforts [6]. Conversely, nations with lower PC1 scores, reflective of higher levels of socio-economic development, consistently achieve greater vaccination rates, supported by robust healthcare systems, greater resources, and more widespread educational attainment [6]. However, a few exceptions exist where countries with relatively high PC1 scores still achieve high immunization coverage. While the specific mechanisms driving this phenomenon remain unclear and warrant further investigation, several factors may contribute. These include effective government-led vaccination initiatives, strong community engagement, integration of immunization programs into primary healthcare services, and external support from international and non-governmental organizations. The identified socio-economic development patterns may also be relevant for adult vaccination programs, which often face similar access-related barriers, such as healthcare infrastructure limitations, public trust, and socio-economic inequities [8]. Future research should explore the applicability of PCA-derived development scores to adult immunization datasets, which could help identify cross-cutting structural barriers and inform more inclusive vaccination strategies.

Overall, these findings align with prior research on global health disparities [4,5], illustrating the entrenched structural inequalities in healthcare infrastructure, economic resources, and social determinants of health that perpetuate differences in immunization coverage. The study underscores the value of focusing on core dimensions of development, as captured by the PC1 score, while acknowledging the broader socio-economic factors that complement these dominant drivers. This approach offers a detailed perspective on how disparities in development contribute to inequities in public health outcomes. By integrating socio-economic, infrastructural, and health-related dimensions, this framework has the potential to inform evidence-based interventions, optimize resource allocation, and support the realization of global equity objectives [37], such as those outlined in the Immunization Agenda 2030 [10]. Furthermore, it serves as a flexible and scalable tool for examining other public health challenges, ensuring that efforts to address health disparities are grounded in a thorough understanding of the complex socio-economic factors that shape health outcomes.

From a methodological point of view, the framework we propose aligns with alternative approaches commonly used to study immunization inequality, such as multi-level modeling [38], which accounts for hierarchical data structures and concentration indices, which quantify inequality along socio-economic gradients. Additionally, stratification methods based on wealth quintiles, maternal education, or urban–rural residence are frequently employed in demographic and health surveys [39]. While our PCA-based approach emphasizes macro-level structural disparities, these complementary methods provide valuable insights, particularly when individual- or household-level data are available.

This study presents several strengths and limitations that warrant consideration. Among its strengths, the use of PCA-derived composite scores offers a sophisticated framework for analyzing disparities in immunization coverage. By integrating a wide array of socio-economic, health-related, and infrastructural indicators, this approach transcends traditional metrics, such as GDP per capita, providing a more nuanced understanding of the complex drivers of vaccination coverage. The application of these scores to both historical (2000–2015) and contemporary (2016–2021) data enhances their value for longitudinal and comparative analyses. The long timeframe of this study provides a unique opportunity to examine how the socio-economic determinants of immunization may evolve over time. While our PCA approach showed strong stability between the training (2000–2015) and testing (2016–2021) periods—with high correlations between component structures—some degree of temporal drift in underlying development factors is expected. A more granular temporal analysis could further suggest how the drivers of immunization disparities shift over time and guide the design of time-sensitive policy responses. This represents a valuable avenue for future research. Moreover, the study’s global scope, encompassing 195 countries, ensures a comprehensive evaluation of disparities, enabling insights into regional and temporal trends that are crucial for informing global health policies.

However, the study is not without limitations. The reliance on country-level data masks significant within-country inequalities, particularly in geographically and socio-economically diverse nations. In large, socioeconomically heterogeneous countries, intra-national variation in immunization coverage can be as pronounced as inter-country differences. Such disparities are frequently driven by factors, including urban–rural divides, uneven distribution of health infrastructure, and region-specific governance challenges [6,40,41]. Aggregating data at the national level may obscure critical inequities experienced by marginalized sub-populations—particularly those in rural areas or belonging to lower socio-economic strata—who are disproportionately affected by under-vaccination. To enhance the granularity, accuracy, and policy relevance of future analyses, a more systematic and sustained effort is needed to collect, harmonize, and integrate sub-national immunization data. This includes leveraging regional health surveys, district-level immunization records, and routine administrative data where available. Furthermore, coupling these sub-national indicators with spatial analysis methods could significantly improve the applicability of PCA-derived scores for informing geographically targeted health interventions. Investing in robust sub-national data systems should be a priority for both researchers and policymakers seeking to uncover and address immunization inequities that remain hidden at the national scale. Moreover, the use of the same countries in both the training and testing datasets across different time periods may introduce country-level dependency, potentially inflating model performance metrics by preserving structural similarities between the two datasets. Another limitation lies in the quality and completeness of the input data. Although robust data management protocols were employed, including imputation techniques for missing values, the exclusion of certain variables due to high levels of missing data may have influenced the findings. To address missing data, we employed a combination of forward and backward imputation methods. These techniques, while computationally efficient and based on the assumption of temporal continuity, do not leverage inter-variable correlations and may not fully capture the dynamics of rapidly changing socio-economic indicators. However, such abrupt changes are relatively rare in the dataset, and the overall proportion of missing values is low. To further assess the potential impact of this imputation strategy, we conducted a sensitivity analysis using a complete-case dataset. The consistency of key associations across both imputed and non-imputed datasets supports the robustness of our main findings with respect to missing data handling. A further important limitation of this study is the omission of cultural, behavioral, and psychological determinants of vaccine uptake [42]—most notably, vaccine hesitancy. While the present PCA-based framework was designed to capture structural socio-economic factors, non-structural influences, such as trust in healthcare systems, exposure to misinformation, religious or ideological beliefs, and political attitudes, can play a substantial role in shaping immunization behaviors [43,44,45,46]. In contexts where structural conditions are favorable (e.g., high-income settings), the influence of behavioral and cultural determinants, such as vaccine hesitancy, might override structural advantages, potentially limiting vaccine uptake despite favorable socio-economic environments. Conversely, in lower-income settings or regions facing infrastructural challenges, behavioral determinants might exacerbate existing barriers, intensifying disparities in vaccine coverage [43,44,45,46]. Given the inherently complex and context-specific nature of these interactions, conventional development indicators alone may be insufficient. Future research should consider employing mixed-methods approaches, integrating qualitative insights and digital trace data, or extending PCA models to incorporate behavioral variables. This would allow a more nuanced understanding of the interplay between structural and non-structural determinants in shaping immunization behaviors across diverse contexts.

## 5. Conclusions

This study demonstrates the utility of PCA-derived socio-economic development scores in understanding and addressing global health disparities. By integrating diverse indicators into a single framework, the scores provide a powerful tool for identifying regions and populations most in need of intervention. Addressing upstream determinants, such as education, healthcare infrastructure, and governance, is essential for reducing inequities and achieving global immunization targets. These findings underscore the importance of multi-dimensional approaches to health policy and the need for continued efforts to ensure equitable access to life-saving vaccines. Future research should aim to address limitations by integrating sub-national data, expanding the range of variables to include cultural and behavioral determinants, and refining the methodology to better capture the complex, multi-faceted nature of immunization inequities. Such advancements would not only strengthen the validity of the findings but also enhance the utility of PCA-derived scores as tools for guiding global and regional health policies. By building on this work, future studies can contribute to a deeper understanding of the socio-economic factors driving health disparities and support more effective strategies for achieving equitable immunization coverage worldwide.

## Figures and Tables

**Figure 1 ijerph-22-00941-f001:**
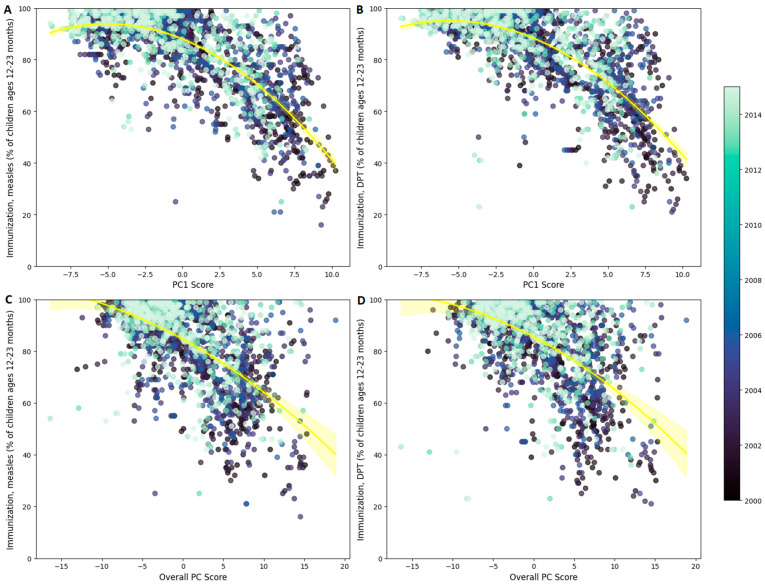
Scatter plots of the relationship between development scores and immunization coverages. Panels (**A**,**B**) illustrate the relationship between the PC1 Score and immunization coverage for measles and DTP, respectively, while Panels (**C**,**D**) present the association between the Overall PC Score and immunization coverage for measles and DTP. The yellow polynomial trend line—with its shaded 95% confidence interval—depicts the overall fitted relationship across all observations. Each dot represents a country-year observation and is color-coded by year (from 2000 to 2015).

**Figure 2 ijerph-22-00941-f002:**
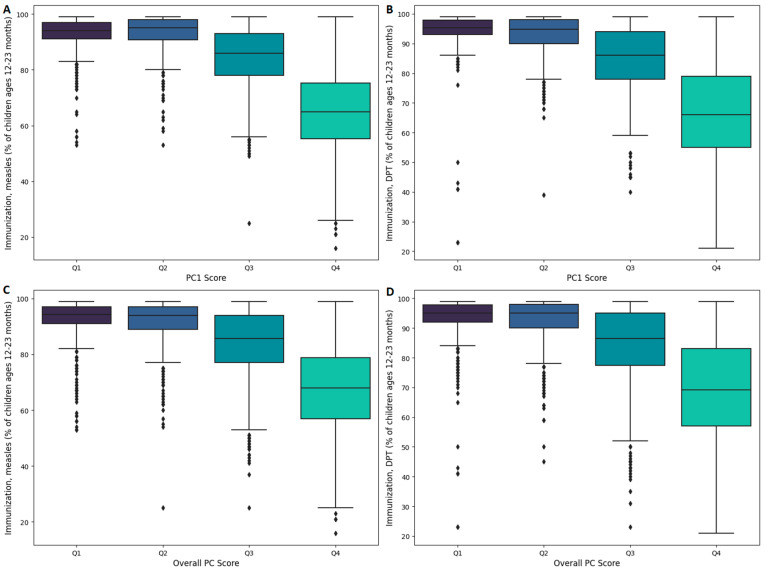
Box plots illustrating the relationship between socio-economic development quartiles and immunization coverage for measles (**A**,**C**) and DTP (**B**,**D**). Panels (**A**,**B**) display the data based on PC1 Score quartiles, while Panels (**C**,**D**) show the corresponding relationships using Overall PC Score quartiles. In every panel, Q1 corresponds to countries with the highest socio-economic development, with development levels progressively decreasing through Q2 and Q3 to Q4, which includes the least developed countries. Within each box plot, the horizontal line represents the median immunization coverage. The box itself spans the interquartile range (IQR), covering the central 50% of the observations from the 25th to the 75th percentile. The whiskers extend to the most extreme values within 1.5 times the IQR, and any observations falling outside this range are plotted individually as outliers.

**Figure 3 ijerph-22-00941-f003:**
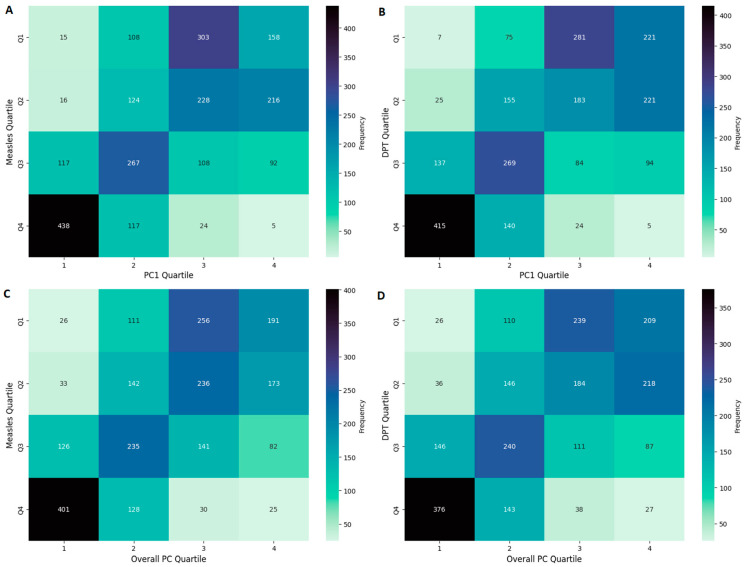
Confusion matrices illustrating the agreement between quartiles of PCA-based development scores and immunization coverage for measles (**A**,**C**) and DTP (**B**,**D**). Panels (**A**,**B**) correspond to the PC1 Score, whereas panels (**C**,**D**) show results for the Overall PC Score. Each matrix cell indicates the frequency of observations (i.e., the number of country-year observations) assigned to each combination of development score quartile and immunization coverage quartile. Darker shades represent higher frequencies. The diagonal runs from the top-right cell to the bottom-left cell, indicating concordance between a country’s development score classification and its immunization coverage quartile.

**Figure 4 ijerph-22-00941-f004:**
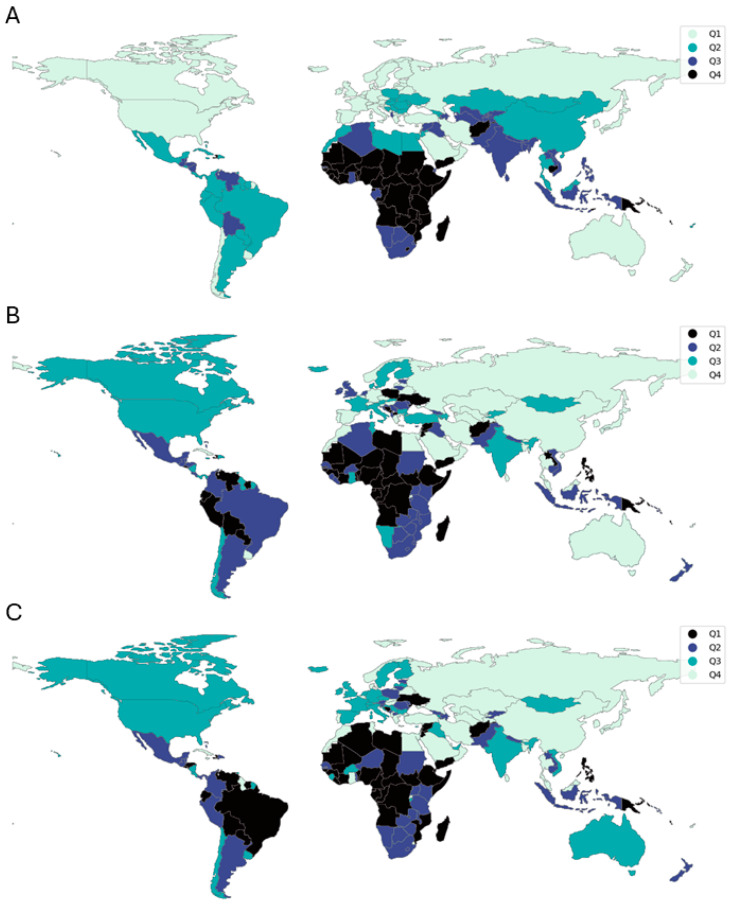
Quartile maps showing the geographical distribution of PC1 score (**A**) and immunization coverage for measles (**B**) and DTP (**C**) in 2021.

## Data Availability

The data presented in this study are available from the World Bank, Data Bank: World Development Indicators (https://databank.worldbank.org/reports.aspx?source=2&country=ARE; accessed on 1 February 2025) and UNICEF Data Warehouse (https://data.unicef.org/resources/data_explorer/unicef_f/; accessed on 1 February 2025), reference number [19] and [20], respectively.

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
