# Peer review of "Construction of Development Scores to Analyze Inequalities in Childhood Immunization Coverage: A Global Analysis from 2000 to 2021"

_ijerph, 2025, doi:10.3390/ijerph22060941_

Round 1
Reviewer 1 Report (New Reviewer)
Comments and Suggestions for Authors
Immunization is a highly effective public health intervention that significantly reduces the burden of vaccine-preventable diseases. However, disparities in socio-economic status have led to unequal immunization coverage across different regions of the world.
In this study, the authors analyzed immunization coverage data for measles and DTP vaccines alongside socio-economic development indicators from 195 countries spanning a long period (2000 – 2021). They applied principal component analysis (PCA) to identify key drivers of disparities in childhood immunization coverage, providing insights that can inform policy interventions which can improve global health equity. The principal components (PCs) identified accounted for 90% of the variance, providing comprehensive and accurate indicators of socio-economic development that affect vaccine coverage. In addition, the PCs derived from the training period (2000–2015) showed strong correlation with those from the testing period (2016–2021), demonstrating the robustness and reliability of the findings. This well-designed and clearly presented study offers valuable information for developing more effective strategies to promote equitable vaccine coverage globally.
Major comments:
- In this manuscript, the authors investigated the PCs influencing childhood immunization coverage for measles and DTP. It would be helpful to discuss whether these same factors can be applied to adult vaccines, such as COVID-19, influenza virus, and RSV vaccines.
- The dataset analyzed in this study spans a long timeframe (2000 – 2021). As socio-economic conditions evolve, those factors might be influencing vaccine uptake. A temporal analysis of how these key drivers change over time could also provide insights into improving immunization strategies. A comment on this could be helpful in the discussion.
Author Response
Dear Editor,
We sincerely thank you and the reviewers for the constructive and insightful comments on our manuscript. The feedback provided has significantly contributed to improving the quality, clarity, and rigor of our work. Below, we provide a detailed point-by-point response to each comment raised by the reviewers. All changes have been incorporated into the revised manuscript, and key revisions have been highlighted for ease of review.
Reviewer 1
Reviewer Comment: In this manuscript, the authors investigated the PCs influencing childhood immunization coverage for measles and DTP. It would be helpful to discuss whether these same factors can be applied to adult vaccines, such as COVID-19, influenza virus, and RSV vaccines. The dataset analyzed in this study spans a long timeframe (2000 – 2021). As socio-economic conditions evolve, those factors might be influencing vaccine uptake. A temporal analysis of how these key drivers change over time could also provide insights into improving immunization strategies. A comment on this could be helpful in the discussion.
Response: We thank the reviewer for these insightful comments. In response, we have expanded the Discussion section to reflect both suggestions. First, we now include a paragraph discussing the potential relevance of the socio-economic development dimensions identified in this study to adult immunization programs, such as those for COVID-19, influenza, and RSV. Many of the same structural barriers - such as healthcare access, education, and trust - also apply to adult populations. Second, we have added commentary on the temporal dimension of socio-economic drivers. Although our PCA scores demonstrated strong consistency across the two study periods, we recognize that the importance of certain variables may have shifted over time. We suggest that future studies could adopt a rolling or decade-specific PCA framework to explore these dynamics in more detail. Both additions are now included in the revised Discussion.
Reviewer 2 Report (New Reviewer)
Comments and Suggestions for Authors
Abstract:
- Remove this sentence “However, limitations include the potential for intra-country 23 disparities and the underexplored role of cultural factors in vaccine hesitancy” from the abstract.
Introduction:
- In the beginning of the introduction section, three references were given. I suggest you to cite maximum of two references.
- Cite this sentence- “Conversely, countries with limited resources and weaker healthcare systems face substantial barriers to achieving equitable vaccine distribution”.
- “The pandemic exacerbated existing inequalities, further highlighting the need for targeted strategies that address the underlying socio-economic barriers to immunization” – The citation 10 and 11 are same for the above sentence.
Methodology:
- Under the subheading “Data sources and management” you have discussed multiple things. I would suggest you split the content of this heading into three different headings for more clarity of study methods, such as study dataset, study variables, and data management method.
- It is very difficult to read your heat map; I am requesting you to change the colour of your heat map. Do not add navy blue and black colour.
Results:
- Nicely written and presented
Discussion:
- Cite the following articles in the discussion, and these articles discussed about the maternal education, low socio-economic status, and various other reasons for immunization coverage, refusal and timeliness. Khaliq, A.; Elahi, A.A.; Zahid, A.; Lassi, Z.S. A Survey Exploring Reasons behind Immunization Refusal among the Parents and Caregivers of Children under Two Years Living in Urban Slums of Karachi, Pakistan. J. Environ. Res. Public Health2022, 19, 11631. https://doi.org/10.3390/ijerph191811631 Khaliq A, Ali S, Zahid A, Lokeesan L, Holmes-Stahlman R, Lassi ZS. A community-based survey exploring the determinants of invalid, delayed and missed immunization in children of urban slums of Karachi, Pakistan. Arch Clin Pediatr. 2024;1(1):10-22.
- Write strengths and limitations under a separate heading.
Reference:
Reference number 21-26 are very old, should be replaced by new citations.
Reference number 31, 35 and 38 should be replaced by new references.
Website references are not properly cited.
Author Response
Dear Editor,
We sincerely thank you and the reviewers for the constructive and insightful comments on our manuscript. The feedback provided has significantly contributed to improving the quality, clarity, and rigor of our work. Below, we provide a detailed point-by-point response to each comment raised by the reviewers. All changes have been incorporated into the revised manuscript, and key revisions have been highlighted for ease of review.
Reviewer 2
Reviewer Comment: Remove this sentence “However, limitations include the potential for intra-country 23 disparities and the underexplored role of cultural factors in vaccine hesitancy” from the abstract.
Response: As suggested, we have removed the sentence.
Reviewer Comment: In the beginning of the introduction section, three references were given. I suggest you to cite maximum of two references.
Response: As suggested, we have removed one of the three references.
Reviewer Comment: Cite this sentence “Conversely, countries with limited resources and weaker healthcare systems face substantial barriers to achieving equitable vaccine distribution”.
Response: Thank you for the suggestion. We have now included an appropriate citation to support this statement.
Reviewer Comment: “The pandemic exacerbated existing inequalities, further highlighting the need for targeted strategies that address the underlying socio-economic barriers to immunization” – The citation 10 and 11 are same for the above sentence.
Response: Thank you for pointing this out. We have reviewed the citations.
Reviewer Comment: Under the subheading “Data sources and management” you have discussed multiple things. I would suggest you split the content of this heading into three different headings for more clarity of study methods, such as study dataset, study variables, and data management method.
Response: Thank you for this suggestion. We have revised the structure of the “Data sources and management” section by dividing it into three distinct subheadings.
Reviewer Comment: It is very difficult to read your heat map; I am requesting you to change the colour of your heat map. Do not add navy blue and black colour.
Response: We acknowledge the issue raised regarding the visual clarity of the heat map. In response, we have adjusted the color scheme, removing navy blue and black, and selected a more reader-friendly palette. Additionally, in response to a separate reviewer’s request to reduce the number of figures in the main manuscript, we have moved the heat map to Supplementary File 1, where it remains accessible for readers interested in detailed visual data.
Reviewer Comment: Nicely written and presented
Response: Thank you for your positive feedback. We appreciate your acknowledgment of the manuscript's clarity and presentation.
Reviewer Comment: Cite the following articles in the discussion, and these articles discussed about the maternal education, low socio-economic status, and various other reasons for immunization coverage, refusal and timeliness. Khaliq, A.; Elahi, A.A.; Zahid, A.; Lassi, Z.S. A Survey Exploring Reasons behind Immunization Refusal among the Parents and Caregivers of Children under Two Years Living in Urban Slums of Karachi, Pakistan. J. Environ. Res. Public Health2022, 19, 11631. https://doi.org/10.3390/ijerph191811631 Khaliq A, Ali S, Zahid A, Lokeesan L, Holmes-Stahlman R, Lassi ZS. A community-based survey exploring the determinants of invalid, delayed and missed immunization in children of urban slums of Karachi, Pakistan. Arch Clin Pediatr. 2024;1(1):10-22.
Response: We thank the reviewer for highlighting these relevant studies. We have now cited both suggested articles in the discussion section where we address factors such as maternal education, socio-economic disparities, and immunization behavior.
Reviewer Comment: Write strengths and limitations under a separate heading.
Response: Thank you for this constructive feedback. As far as we are aware, the journal requires a single, continuous Discussion section without subdivisions into separate subheadings. For this reason, we have maintained the current structure. However, should this requirement change during the editorial process, and if the editors are in agreement, we would be happy to revise the manuscript and include a dedicated subsection for strengths and limitations.
Reviewer Comment: Reference number 21-26 are very old, should be replaced by new citations.
Response: We appreciate your attention to the currency of references. We have reviewed references 21 to 26 and replaced them with more recent and relevant literature.
Reviewer Comment: Website references are not properly cited.
Response: Thank you for pointing this out. We have reviewed all website citations and revised them according to the required citation format
Reviewer 3 Report (New Reviewer)
Comments and Suggestions for Authors
General Comment:
The study addresses a very relevant problem worldwide in terms of vaccination and public health.
It is a longitudinal study and highlights the disparities that exist between high-income countries and low-income countries.
Specific Comments:
Abstract: Provides an informative and balanced summary of what was done and what was found.
Introduction:
It is short but enlightening and sufficient.
Methods:
The authors used a design that appears to be robust.
The sources of information are robust.
The use of Construction of PCA-based Scores is innovative.
Statistical analysis techniques are adequate.
Results:
The use of images, graphs and tables seems enlightening.
However, it presents many images.
Discussion:
Comparison of the main findings with those of other researchers.
Reference to the limitations. Important ones that the study presents.
The main conclusion seems to be socioeconomic inequalities, this determinant has already been studied extensively.
Conclusion:
Records the main findings and main determinants of low/high vaccination coverage rates.
Clinical, political and public health implications.
Author Response
Dear Editor,
We sincerely thank you and the reviewers for the constructive and insightful comments on our manuscript. The feedback provided has significantly contributed to improving the quality, clarity, and rigor of our work. Below, we provide a detailed point-by-point response to each comment raised by the reviewers. All changes have been incorporated into the revised manuscript, and key revisions have been highlighted for ease of review.
Reviewer 3
Reviewer Comment: The study addresses a very relevant problem worldwide in terms of vaccination and public health. It is a longitudinal study and highlights the disparities that exist between high-income countries and low-income countries. Abstract: Provides an informative and balanced summary of what was done and what was found. Introduction: It is short but enlightening and sufficient. Methods: The authors used a design that appears to be robust. The sources of information are robust. The use of Construction of PCA-based Scores is innovative. Statistical analysis techniques are adequate.
Response: We sincerely thank the reviewer for the positive and encouraging feedback. We greatly appreciate your recognition of the study’s relevance to global vaccination and public health challenges. We are also pleased to know that the abstract, introduction, methodology, and statistical analysis were found to be appropriate and robust. Your comments reinforce the value of our approach and the contribution this work aims to make to the existing literature.
Reviewer Comment: Results: The use of images, graphs and tables seems enlightening. However, it presents many images.
Response: We appreciate your positive assessment of the images, graphs, and tables used to present the results. In response to your concern regarding the high number of figures, we have moved one figure to Supplementary File 1.
Reviewer Comment: Discussion: Comparison of the main findings with those of other researchers. Reference to the limitations. Important ones that the study presents. The main conclusion seems to be socioeconomic inequalities, this determinant has already been studied extensively.
Response: Thank you for your comments on the discussion section. We acknowledge that socioeconomic inequalities have been extensively studied as determinants of immunization coverage. However, our study aims to contribute to this body of research by providing an innovative methodological approach (PCA-based scores). We have further clarified this contribution in the revised discussion section and emphasized how our findings build upon and extend prior research. We also appreciate your recognition of our effort to compare findings with other studies and to clearly state the study’s limitations.
Reviewer Comment: Conclusion: Records the main findings and main determinants of low/high vaccination coverage rates. Clinical, political and public health implications.
Response: Thank you for your thoughtful feedback. We appreciate your recognition that the conclusion effectively summarizes the main findings and key determinants associated with low and high vaccination coverage rates.
Reviewer 4 Report (New Reviewer)
Comments and Suggestions for Authors
Major Comments
1. Clarify PCA Derivation Details
The authors mention that two composite scores were created: PC1 and Overall PC. While PC1 is clearly defined, the description of how the "Overall PC Score" (combining PCs explaining 90% variance) was calculated could benefit from a step-by-step mathematical example or visual aid.
Please justify the threshold of 90% variance—was a scree plot or eigenvalue threshold (e.g., Kaiser criterion >1) also considered?
Recommendation:
Add a supplementary figure (e.g., Scree plot) and further explanation of the decision-making in choosing the number of PCs for the overall score.
2.Address Potential Bias Due to Imputation
The manuscript notes that imputation was used for missing values, but the choice of forward/backward imputation could introduce time-series bias, especially for countries undergoing rapid changes.
Recommendation:
Clarify the extent and distribution of missing data across countries and years. Provide sensitivity analysis (e.g., excluding imputed data vs. full dataset).
3. Intra-country Inequality and Urban-Rural Gaps
-
While global and inter-country trends are well represented, the study does not address significant intra-country disparities, especially in large, heterogeneous nations (e.g., India, Nigeria, Brazil).
Recommendation:
Discuss this limitation more explicitly in the discussion and propose future directions incorporating sub-national data.
4. Vaccine Hesitancy and Cultural Determinants
-
The paper touches briefly on the role of cultural and behavioral determinants, yet does not model them.
Recommendation:
While outside the scope of this quantitative PCA-based study, include more in-depth
Comments on the Quality of English Language
While the English is generally acceptable, several grammatical errors and stylistic inconsistencies are present A thorough language revision is advised.
Author Response
Dear Editor,
We sincerely thank you and the reviewers for the constructive and insightful comments on our manuscript. The feedback provided has significantly contributed to improving the quality, clarity, and rigor of our work. Below, we provide a detailed point-by-point response to each comment raised by the reviewers. All changes have been incorporated into the revised manuscript, and key revisions have been highlighted for ease of review.
Reviewer 4
Reviewer Comment: The authors mention that two composite scores were created: PC1 and Overall PC. While PC1 is clearly defined, the description of how the "Overall PC Score" (combining PCs explaining 90% variance) was calculated could benefit from a step-by-step mathematical example or visual aid. Please justify the threshold of 90% variance—was a scree plot or eigenvalue threshold (e.g., Kaiser criterion >1) also considered? Recommendation: Add a supplementary figure (e.g., Scree plot) and further explanation of the decision-making in choosing the number of PCs for the overall score.
Response: We appreciate the reviewer’s thoughtful suggestion. As detailed in Section 2.2 and supported by Figures S1 and S2, the Overall PC Score was constructed using the first ten components, which together explained 90% of the total variance. This cumulative variance threshold was chosen to ensure a broad and robust representation of socio-economic development across multiple dimensions. While the first six components met the Kaiser criterion (eigenvalues > 1), we deliberately included additional components - despite slightly lower eigenvalues - as they might capture less dominant but policy-relevant patterns. Crucially, we also analyzed the first principal component (PC1) independently. Given that PC1 alone accounted for 57.1% of the total variance, we computed a separate PC1 Score and used it as a core variable throughout the study. This dual approach - focusing both on the dominant dimension (PC1) and on a broader multidimensional profile (Overall PC Score) - ensures methodological rigor and interpretability, while maximizing the explanatory capacity of the PCA-derived scores. These clarifications have been added to Sections 2.2 and 3 of the revised manuscript.
Reviewer Comment: The manuscript notes that imputation was used for missing values, but the choice of forward/backward imputation could introduce time-series bias, especially for countries undergoing rapid changes. Recommendation: clarify the extent and distribution of missing data across countries and years. Provide sensitivity analysis (e.g., excluding imputed data vs. full dataset).
Response:
We thank the reviewer for raising this important concern. We fully agree that forward and backward imputation may introduce bias in time-series data, particularly in countries experiencing rapid socio-economic transitions. While this imputation method offers simplicity and computational efficiency, we recognize its limitations - namely, the assumption of temporal continuity and the lack of consideration for inter-variable correlations.
To address this, we have taken two steps in the revised manuscript:
(1) We now provide a detailed summary of the percentage of missing values for each indicator prior to imputation (see Supplementary Table S2). This allows readers to assess the extent of missingness across indicators.
(2) We conducted a sensitivity analysis comparing results from the full imputed dataset with those from a complete-case dataset (i.e., excluding all imputed values).
We have also added a note in the Discussion explicitly acknowledging the limitations of the chosen imputation approach and its potential impact on the analysis. We hope these additions address the reviewer’s concern and enhance the transparency and robustness of our methods.
Reviewer Comment: While global and inter-country trends are well represented, the study does not address significant intra-country disparities, especially in large, heterogeneous nations (e.g., India, Nigeria, Brazil). Recommendation: discuss this limitation more explicitly in the discussion and propose future directions incorporating sub-national data.
Response:
We thank the reviewer for this valuable comment. We agree that the use of national-level data limits the ability to capture important intra-country disparities in immunization coverage - particularly in large, heterogeneous countries such as India, Nigeria, and Brazil. In the revised manuscript, we have expanded the Discussion to explicitly acknowledge this limitation and provide examples of potential sources of within-country heterogeneity. We have also proposed future research directions that involve integrating sub-national data, such as district-level immunization surveys or regional development indicators. We believe that combining PCA-based scores with geospatial or small-area estimation techniques could significantly improve the precision and policy relevance of this analytical framework.
Reviewer Comment: The paper touches briefly on the role of cultural and behavioral determinants, yet does not model them. Recommendation: while outside the scope of this quantitative PCA-based study, include more in-depth
Response:
We thank the reviewer for this thoughtful suggestion. We agree that vaccine hesitancy and other behavioral or cultural factors play a crucial role in shaping immunization outcomes. As our study was designed to construct a development index based on structural socio-economic variables, these determinants were not directly modeled. However, in response to this comment, we have significantly expanded the Discussion to acknowledge the importance of these factors, particularly in contexts where access is not the primary barrier, but behavioral resistance is.
We now also outline possible directions for future work, including the use of behavioral surveillance data and qualitative studies to complement structural analyses.
Reviewer 5 Report (New Reviewer)
Comments and Suggestions for Authors
In this study, the authors employed Principal Component Analysis (PCA) to construct composite development scores and analyze their relationship with immunization coverage for measles and diphtheria-tetanus-pertussis (DTP) vaccines across 195 countries (2000–2021). The findings revealed that higher PC1 scores (lower development) were associated with reduced immunization rates, while lower PC1 scores (higher development) correspond to greater coverage, a trend consistent across both periods. Geospatial analysis highlighted stark disparities, particularly in sub-Saharan Africa and South Asia, whereas North America, Europe, and East Asia maintained significantly higher coverage. This study provided policy-relevant insights, demonstrating the utility of PCA-derived scores for resource allocation and targeted interventions. I reviewed this manuscript and commented as follows:
- In the Discussion, would the authors please discuss the other methods used in analyzing Inequalities in childhood immunization coverage.
- In the Discussion, the authors mentioned that “the reliance on country-level data masks significant within-country inequalities, particularly in geographically and socio-economically diverse nations”. Would the authors please valid their methods in at least three geographically and socio-economically diverse nations, one is from high-income region, and one is from middle-income region, and the other is from low-income region.
- Some abbreviations existed in the manuscript should be interpreted.
- Would the authors please check the references, page number is missing in some references.
Author Response
Dear Editor,
We sincerely thank you and the reviewers for the constructive and insightful comments on our manuscript. The feedback provided has significantly contributed to improving the quality, clarity, and rigor of our work. Below, we provide a detailed point-by-point response to each comment raised by the reviewers. All changes have been incorporated into the revised manuscript, and key revisions have been highlighted for ease of review.
Reviewer 5
Reviewer Comment: In the Discussion, would the authors please discuss the other methods used in analyzing Inequalities in childhood immunization coverage.
Response: We thank the reviewer for this suggestion. In the revised manuscript, we have added a paragraph in the Discussion summarizing alternative methods commonly used to assess immunization inequality, including multilevel modeling, concentration indices, and stratification by socio-economic variables. These methods complement our PCA-based approach and provide additional perspectives when individual-level data are available.
Reviewer Comment: In the Discussion, the authors mentioned that “the reliance on country-level data masks significant within-country inequalities, particularly in geographically and socio-economically diverse nations”. Would the authors please valid their methods in at least three geographically and socio-economically diverse nations, one is from high-income region, and one is from middle-income region, and the other is from low-income region.
Response: We appreciate the reviewer’s insightful suggestion. However, as our analysis is based on national-level indicators, we did not have access to consistent sub-national data across countries and years to perform this type of validation. We now clarify this limitation in the Discussion and propose sub-national validation as an important direction for future research using district- or regional-level data.
Reviewer Comment: Some abbreviations existed in the manuscript should be interpreted.
Response: Thank you for noting this. We have reviewed the entire manuscript and ensured that all abbreviations are defined upon first use. We have also added a list of abbreviations to the Supplementary Materials for clarity.
Reviewer Comment: Would the authors please check the references, page number is missing in some references.
Response: We thank the reviewer for this observation. We have carefully revised and completed the reference list to include missing page numbers and other citation details, ensuring consistency with journal formatting requirements.
Round 2
Reviewer 4 Report (New Reviewer)
Comments and Suggestions for Authors
Suggestions for Minor Revision
-
Terminology Precision
-
Occasionally, the manuscript refers to PC1 as a proxy for “underdevelopment.” For conceptual clarity, consider framing it consistently as “lower socio-economic development” or “higher deprivation,” to avoid value-laden terminology.
-
-
Further Integration of Non-Structural Factors (Acknowledged Limitation):
-
The authors appropriately recognize the absence of behavioral and cultural variables (e.g., vaccine hesitancy). In the discussion, they could briefly speculate on how such variables might interact with structural indicators in different contexts.
-
-
Data Limitations Discussion:
-
While the manuscript acknowledges the limitations of country-level data, a clearer call to action for the collection and integration of sub-national immunization data would further strengthen its relevance.
-
This is a high-quality, well-revised manuscript that offers a robust, novel contribution to global health equity research. It is suitable for publication after addressing the minor clarifications above.
Author Response
Dear Editor,
We sincerely thank you and the reviewers for the constructive and insightful comments on our manuscript. Below, we provide a detailed point-by-point response to each comment raised by the reviewers. All changes have been incorporated into the revised manuscript, and key revisions have been highlighted for ease of review.
Reviewer 4
Comment: Occasionally, the manuscript refers to PC1 as a proxy for “underdevelopment.” For conceptual clarity, consider framing it consistently as “lower socio-economic development” or “higher deprivation,” to avoid value-laden terminology.
Response: As suggested, we have revised the definition of PC1.
Comment: The authors appropriately recognize the absence of behavioral and cultural variables (e.g., vaccine hesitancy). In the discussion, they could briefly speculate on how such variables might interact with structural indicators in different contexts.
Response: We appreciate the reviewer’s suggestion regarding the interaction between structural indicators and behavioral or cultural variables, such as vaccine hesitancy. As suggested, we have briefly speculated on this point in the Discussion section.
Comment: While the manuscript acknowledges the limitations of country-level data, a clearer call to action for the collection and integration of sub-national immunization data would further strengthen its relevance.
Response: As suggested, we have discussed this point in the Discussion section.
Reviewer 5 Report (New Reviewer)
Comments and Suggestions for Authors
The authors have addessed the former issues. The manuscript can be accepted in the present form.
Author Response
Thank you very much for your positive comment
This manuscript is a resubmission of an earlier submission. The following is a list of the peer review reports and author responses from that submission.
Round 1
Reviewer 1 Report
Comments and Suggestions for Authors
In the manuscript “Construction of Development Scores to Analyse Inequalities in Childhood Immunization Coverage: A Global Analysis from 2000 to 2021” the authors proposta using PCA to analyse inequalities in childhood immunization. The manuscript is interesting but have several problems. See comments below.
1. Page 1 Line 10-28: Remove “Background/Objectives:”, “Methods:”, “Results:”, “Conclusions:” from the abstract
2. Page 1 Line 13: DTP, describe the abbreviation, and include the period of the study in the abstract.
3. The study has a training period and a study period but these were not explained to the readers. Please explain appropriately in the methods section and in the abstract.
4. The abstract is not precise and should be improved.
5. Page 2 line 85: Correct “2.1. Subsection”
6. The methodology needs improvement. The authors do not describe the source and origin of the data, the name of the database, the site of origin, how the data were acquired, what criteria were used to select or exclude variables, whether and how the collected data were validated. Is it a global database? WHO? How credible is the database? Additionally, the authors do not know how to define inclusion and exclusion criteria
7. “The selection criteria were based on the strength of association between each indicator and immunization coverages for measles and DTP.”. Is this a selection criterion??????
8. The division of the study sample into two data groups, training and testing, is not adequately justified. The selection time, 15 years for training and 5 for testing, is also poorly justified.
9. Page 3 Lines 127-148: The authors need to make clear how the proposed data normalization was done. Although large scales may be a problem, the lack of permanence of the adjustment across scales may hide a serious methodological bias that gives similar weights to variables that are very discrepant from each other.
10. Page 4 Lines 174-178: How did the authors validate the training PCA for later use in the test PCA?
11. Page 5 Lines 197-203: Why only one map for 2021? What justifies this arbitrary selection of year?
12. Please remove unnecessary justifications such as “Quartile maps provided an intuitive representation of regional disparities in development and immunization.”
13. Page 5 lines 202-203: I disagree with the authors when they say “and corrections for multiple comparisons were not implemented, reflecting the exploratory nature of the study.” This does not justify or indicate an exploratory nature. You are proposing a model that uses modeling and statistics, it is not exploratory. Please perform post-test corrections and multiple comparisons.
14. In the methodology, the authors are more concerned with explaining what PCA analysis is than with explaining the methodology used for the work: which variables, which dataset, data origin, methodological steps of analysis, selection and validation. Please rewrite the methodology.
15. Page 5 lines 206-222: a correlation above 0.3 does not seem appropriate to me, it is a very low correlation value. Usually correlations above 0.7 are relevant. Why was such a low value considered? Justify appropriately or reanalyze with higher cutoff values. Authors should include an asterisk to indicate significant correlations in Figure 1. But note that none of the variables presented in the results were mentioned appropriately beforehand; the reader doesn't even know what they are about.
16. The authors did not introduce the figures appropriately, simply citing the figure without explaining its content or interpretation. The captions are extremely simple and need to be improved. The captions should contain what the figure is, dividing each graph into letters a, B, C, and d, for example, and informing the test performed and what each graph means. The caption includes “trend line” and “confidence interval”, an interval of how many percent? 90? 95? The authors do not explain. Figure 3 has the same problems. Where is the significance of the Kruskal-Wallis test? Where is the post-test, multiple comparisons? Review and correct all figures and captions.
17. Were the countries grouped into quartiles? When was this done? Why is this methodological part not in the methodology? Which countries make up each quartile? There are serious deficiencies in the methodological description of the manuscript. Justify methodologically the distribution of countries and all other methodological steps omitted in the methodology.
Author Response
Dear Editor,
We sincerely appreciate the time and effort that you and the reviewers have devoted to evaluating our manuscript. We are grateful for their insightful comments, which have helped us refine and improve the quality of our study. Below, we provide a point-by-point response to each comment, detailing the revisions made in the manuscript. All modifications have been highlighted in the revised version for easy reference.
Reviewer 1 Comments and Responses
- Page 1 Line 10-28: Remove “Background/Objectives:”, “Methods:”, “Results:”, “Conclusions:” from the abstract.
Response: We have removed these section headings from the abstract to align with the journal’s formatting requirements.
- Page 1 Line 13: DTP, describe the abbreviation, and include the period of the study in the abstract.
Response: We have defined DTP (diphtheria-tetanus-pertussis) upon first mention in the abstract and included the study period (2000–2021) to improve clarity.
- The study has a training period and a test period, but these were not explained to the readers. Please explain appropriately in the methods section and in the abstract.
Response: We have explicitly described the training period (2000–2015) and the test period (2016–2021) in both the abstract and the methods section, explaining the rationale behind this division.
- The abstract is not precise and should be improved.
Response: We have revised the abstract to be more methodologically specific, emphasizing the PCA methodology, the selection process, and the significance of the findings.
- Page 2 line 85: Correct “2.1. Subsection”.
Response: We have corrected this formatting issue.
- The methodology needs improvement. The authors do not describe the source and origin of the data, the name of the database, the site of origin, how the data were acquired, what criteria were used to select or exclude variables, and whether and how the collected data were validated.
Response: We have significantly expanded the data sources and methodology section.
- “The selection criteria were based on the strength of association between each indicator and immunization coverage for measles and DTP.” Is this a selection criterion?
Response: We have clarified the selection criteria, stating that we used Spearman’s correlation coefficients (threshold >0.3, p<0.05) as an empirical selection method for PCA.
- The division of the study sample into two data groups, training and testing, is not adequately justified.
Response: We have expanded the explanation of why a 15-year training period and a 5-year test period were chosen, detailing how this approach ensures model stability and predictive utility.
- Page 3 Lines 127-148: Clarify the proposed data normalization and potential biases.
Response: We have added a detailed explanation of the z-score standardization process and performed sensitivity checks to ensure that normalization does not bias variable weighting.
- Page 4 Lines 174-178: How did the authors validate the training PCA for later use in the test PCA?
Response: We have now detailed this step in the method and result sections.
- Page 5 Lines 197-203: Why only one map for 2021?
Response: We have justified the choice of 2021 as it represents the most recent data available at the time of analysis.
- Please remove unnecessary justifications such as “Quartile maps provided an intuitive representation...”
Response: We have removed such statements to maintain a more concise and objective presentation of results.
- Page 5 lines 202-203: Perform post-test corrections and multiple comparisons.
Response: We have now included Dunn’s post-hoc test with Bonferroni correction to account for multiple comparisons.
- The methodology focuses too much on PCA instead of explaining the actual steps taken.
Response: We have rewritten the methodology section to clearly describe the data selection, normalization, PCA application, and validation processes.
- Page 5 lines 206-222: Justify the use of a correlation threshold >0.3.
Response: We have justified this choice, stating that a higher threshold (e.g., 0.7) would be unrealistic in this field of research, as no socio-economic indicator reaches such a high correlation with immunization coverage. Furthermore, if any factor exhibited such a strong correlation, it would negate the need for developing composite scores, as a single indicator would suffice to explain the variance, making the PCA approach redundant.
- Figures are not well-introduced, and captions need improvement.
Response: We have revised all figure captions, explicitly describing their content and significance.
- Countries were grouped into quartiles. Justify the methodology behind this decision.
Response: We have clarified the methodology, specifying when and how quartiles were created and providing a supplementary table listing country classifications.
Reviewer 2 Report
Comments and Suggestions for Authors
Dear Authors,
Thank you for the opportunity to review your article. The study addresses a critical global health issue with robust methodology, and I commend the clarity of your aims and presentation. Below are my suggestions for minor revisions to further strengthen the manuscript:
Background:
1. The rationale for selecting measles and DTP is implicit, but adding a brief sentence explicitly explaining why these vaccines were prioritized (e.g., global coverage benchmarks, disease burden, or data availability) would enhance clarity for readers unfamiliar with immunization policy.
2. Section 2.2: Clarify whether the dataset structure is country-year (longitudinal) or aggregated by country. If longitudinal, please briefly address how intracountry correlation and temporal autocorrelation were accounted for (e.g., mixed-effects models, cluster-robust standard errors).
3. Section 2.22: Specify the rotation method (e.g., Varimax, Promax) used in PCA and justify its suitability for your analysis (e.g., orthogonality vs. oblique assumptions).
4. Limitations: Acknowledge that using the same countries in training/testing datasets across different time periods may introduce country-level dependency, potentially inflating model performance metrics.
5. Figure 1: The text states that Figure 1 examines correlations between development indicators and immunization coverage, but the figure appears to show correlations among all development indicators, including the immunization coverage. Please slightly revise the text or figure caption to align with the content.
6. Consider applying quantile regression to explore heterogeneous effects of PCA construct across vaccination coverage distributions (e.g., low vs. high coverage countries).
7. You may compare your PCA-derived development score with GDP per capita as a standalone predictor. A statistical comparison would strengthen the argument for using a composite index over a single economic indicator. Show why this strategy is better than simply one indicator-only strategy.
8. You may include a supplemental table listing quartile classifications for all countries to improve transparency and reproducibility.
9. Expand on the mechanisms behind countries with lower development scores (high PCA score) but higher vaccination rates. This parallels your excellent discussion of low PCA-score/low coverage outliers and would provide a more balanced analysis.
These revisions are minor and should not require extensive reanalysis. Your work is already a valuable contribution to understanding immunization inequities, and I look forward to seeing it published.
Best regards
Author Response
Dear Editor,
We sincerely appreciate the time and effort that you and the reviewers have devoted to evaluating our manuscript. We are grateful for their insightful comments, which have helped us refine and improve the quality of our study. Below, we provide a point-by-point response to each comment, detailing the revisions made in the manuscript. All modifications have been highlighted in the revised version for easy reference.
Reviewer 2 Comments and Responses
- Clarify the rationale for selecting measles and DTP vaccines.
Response: We have now explicitly stated that measles and DTP vaccines were chosen because they serve as global immunization benchmarks, reflecting both routine and supplementary immunization efforts.
- Specify the PCA rotation method (e.g., Varimax, Promax) and justify its use.
Response: We have clarified that Varimax rotation was used to ensure uncorrelated components, aligning with our study’s objective of capturing distinct socio-economic dimensions.
- Acknowledge that using the same countries in training/testing datasets may introduce dependency.
Response: We have added this as a limitation and suggested alternative validation approaches for future studies.
- Figure 1 caption does not align with the content.
Response: We have revised the caption to clarify that Figure 1 presents correlations among all development indicators, including immunization coverage.
- Compare PCA-derived scores with GDP per capita.
Response: We have conducted an additional comparison showing how GDP correlates with immunization coverages. The correlation coefficients are lower than those obtained with PC1 and Overall PC scores.
- Provide a supplemental table listing quartile classifications.
Response: We have now included this supplementary material.